# Health Symptoms Related to Pesticide Use in Farmers and Laborers of Ecological and Conventional Banana Plantations in Ecuador

**DOI:** 10.3390/ijerph18031126

**Published:** 2021-01-27

**Authors:** Hans-Peter Hutter, Michael Poteser, Kathrin Lemmerer, Peter Wallner, Michael Kundi, Hanns Moshammer, Lisbeth Weitensfelder

**Affiliations:** 1Department of Environmental Health, Center for Public Health, Medical University Vienna, 1090 Wien, Austria; hans-peter.hutter@meduniwien.ac.at (H.-P.H.); michael.poteser@meduniwien.ac.at (M.P.); kathrin.lemmerer@meduniwien.ac.at (K.L.); peter.wallner@meduniwien.ac.at (P.W.); Michael.kundi@meduniwien.ac.at (M.K.); Lisbeth.weitensfelder@meduniwien.ac.at (L.W.); 2Department of Hygiene, Medical University of Karakalpakstan, Nukus 230100, Uzbekistan

**Keywords:** banana farming, occupational health, pesticides, pesticide sprayers, ecological farming

## Abstract

Conventional banana farming is pesticide-intensive and leads to high exposure of farmworkers. Ecuador is the world’s biggest exporter of bananas. In this field study in 5 communities in Ecuador, we recorded potentially pesticide-associated subjective health symptoms in farmworkers and compared pesticide users to workers in organic farming. With one exception, symptom rates were always higher in the pesticide-exposed group. Significance was reached in 8 out of 19 investigated symptoms with the highest odds ratios (and smallest *p*-values) for local irritation like skin and eye irritation (OR = 3.58, CI 1.10–11.71, and 4.10, CI 1.37–12.31, respectively) as well as systemic symptoms like dizziness (OR = 4.80, CI 1.55–14.87) and fatigue (OR = 4.96, CI 1.65–14.88). Moreover, gastrointestinal symptoms were reported more frequently by pesticide users: nausea (OR = 7.5, CI 1.77–31.77) and diarrhea (OR = 6.43, CI 1.06–30.00). The majority of farmworkers were not adequately protected from pesticide exposure. For example, only 3 of 31 farmworkers that had used pesticides recently reported using gloves and only 6 reported using masks during active spraying. Improved safety measures and a reduction in pesticide use are necessary to protect the health of banana farmworkers.

## 1. Introduction

Extensive use of pesticides in farming of tropical “cash crops” like bananas is a severe problem in many regions of the world. Due to lack of regulation or enforcement of law, education and control of the safe use of pesticides as well as the application of products that are already phased out and prohibited in industrial countries, farmworkers and (small scale) farmers in affected regions are often confronted with severe health risks that easily remain unnoticed by their target consumer markets [1,2,3,4]. In addition, structural forces shaped by unfair global and national power relations as well as by economic, political, and cultural factors endanger the health and lives of the farmers [5]. Furthermore, personal protection equipment is often not available or simply not used because of the hot climate [6]. Moreover, application equipment may be of poor quality and might provide only limited protection or is difficult and burdensome to use, especially under hot conditions. Poor quality and burdensome use would increase handling errors or risky behaviors. Pesticide labels and hazard descriptions on containers are often only provided in foreign languages which may also hamper the correct use of pesticides [6].

Farmworkers as well as the general populations in the vicinity of the plantations may get exposed by direct handling and application of pesticides, as well as by entering pesticide-treated fields, cleaning and storing equipment, indirect routes, and contamination of water, food or clothing [6,7,8].

Pesticide-related health problems and symptoms are frequent and display a variety of clinical symptoms, including cancer as well as neurological, reproductive, and developmental disorders [9,10,11,12,13,14,15,16].

Consequently, a variety of symptoms is observed in populations exposed to multiple pesticides [17]. An analysis of the causal connections between the exposure to single pesticides and their health effects is often complicated by the use of multiple pesticides or even mixtures [18,19].

Banana farming is a very illustrative sector for studies of pesticide-related health effects, as a large amount of these substances is applied in conventional agriculture. Typically, a significant segment of the local population is occupied in the banana plantations and gets exposed, either directly or indirectly. Accordingly, symptoms of acute and chronic pesticide poisoning in banana workers are reported quite often [20,21,22,23,24,25].

The economic situation in most affected regions results in a lack of regulatory measures and law enforcement regarding pesticide use and is also leading to a reduced awareness and a risk-taking attitude of workers [5,26]. The latter is a consequence of their daily fight for a minimal quality of life rather than knowingly accepting conditions for compromised personal health.

Ecuador is one of the largest banana producers in the world and the biggest exporter of bananas [27,28]. In collaboration with UROCAL, an umbrella organization of small-scale producers in southern Ecuador, and Südwind, an NGO representing the concerns of small farmers and unions of plantation workers, we performed a project in 2015, comprising several studies to obtain information about burden and health outcomes in pesticide use and ecological farmworkers of Ecuador.

Our report [29] was delivered to the UROCAL consortium and used to lobby for legal restrictions of pesticide use in Ecuador after which paraquat was restricted in Ecuador in 2018 [30]. Currently, a European NGO coalition is advocating an export ban of pesticides that are no longer allowed in Europe [31]. Therefore, our findings from 2015 are still relevant for informing policy and the public. That study was also a welcome opportunity for us to confirm our prior findings from the Dominican Republic [32] in a different setting and country.

We already presented our results of a cross-sectional medical survey in Ecuador regarding indicators of genotoxicity in conventional and ecological farming [33] where we clearly demonstrated higher frequencies of genotoxicity and cytotoxicity markers in buccal cells of the pesticide users compared to workers in organic farming. We provide here additional results from the same population, focusing on self-reported acute symptoms in conventional and ecological banana farmworkers.

## 2. Materials and Methods

### 2.1. Study Area and Subjects

The cross-sectional study was performed in five Ecuadorian farming communities (Table 1), located in areas ranging from inland North (Quevedo) to seaside South (Buenavista). A map depicting the geographical location of the study areas can be found in the previous paper [33]. The selection of participants was carried out beforehand with support of several local associations, such as ASTAC (Asociación Sindical de Trabajadores Agricolas Bananeros y Campesinos) and UROCAL (Union Regional des Organizaciones Campesinas del Litoral).

Each participant provided informed consent. The study was approved by the Comité Ético De Investigación, ASTAC, Quevedo, Ecuador (20-06-017). A total of 71 farmworkers were surveyed, of which 34 were engaged in conventional farming (pesticide users) and 37 workers in ecological farming (non-pesticide users, controls). The inclusion criteria were: (1) pesticide use for greater than five years, (2) pesticide use at least three weeks before investigation, and (3) an age greater than 18 years. Three workers from conventional farming said they did not use pesticides and were excluded from analysis (if not specifically stated otherwise). Thus, the final sample size was 68.

### 2.2. Questionnaire and Physical Examination

Structured questionnaires were prepared from standard forms [34,35], supplemented by some symptoms communicated by local health authorities, and adapted to local conditions like pesticide application methods. Questions were either yes/no questions, offering a dichotomous choice, or multiple choice questions, offering several fixed alternatives. The questions were related to socio-demographic features, acute and chronic health symptoms (last 6 months), and indicators of pesticide exposure (pesticides applied and practices, safety measures, aerial spraying, etc.), in both working and housing situation (handling of leftover pesticides, etc.).

The form for the pesticide-exposed group comprised, in total, 39 questions (122 response options) and the forms of the control group (non-pesticide use) included 27 questions (89 response options). The forms were filled with the help of interviewers from the study area that were specifically trained by the research team and were considered trustworthy by the participants.

Participants were registered and a code for anonymization purposes was allocated. Weight, size, and age of the study participants were measured. Then, the questionnaire was filled with support of the interviewer.

### 2.3. Statistical Methods

Data from the questionnaire were analyzed by descriptive statistics. Categorical data for the groups of pesticide users and non-users were compared by Chi-squared test or Fisher’s exact probability test for binomial categories. Quantitative data were compared by Mann–Whitney U-test. Symptoms were additionally analyzed for dependence on age and school education covariates by logistic regression. Unadjusted odds ratios are also presented to indicate the (lack of) confounding by the covariates. Differences in the occurrence of symptoms were determined by Nagelkerke’s pseudo R^2^ for each model used.

*p*-values below 0.05 (5% level) are considered statistically significant, *p*-values below 0.01 (1% level) highly significant.

## 3. Results

### 3.1. Socio-Demographics and Education of the Participants

Regarding most socio-demographic data, no difference was detected in the two groups (Table 2). Only the level of education was significantly higher in the non-pesticide users. While in the group of pesticide users 6 persons reported having had no school education, in the ecological group only one reported no school experience. Accordingly, 14 persons from the ecological group had attended a secondary school and only 6 from the pesticide using group. Most of the participants indicated that their parents were/are also working in agriculture. As expected, a significant difference between the two groups was found in terms of current and lifetime use of pesticides (*p* < 0.001).

### 3.2. Exposure

As explained above, three workers in conventional farming reported not having contact with pesticides. Including these three workers, those in conventional farming reported an average lifetime exposure duration of 11.8 years (std. dev. = 9.8). Duration of exposure of workers in organic farming was significantly shorter (*p* = 0.003) with an average of 4.9 years (8.9). Additional information on recent occupational exposure to pesticides was only obtained from the farmworkers of conventional agriculture (*n* = 31). Spraying within the last three weeks before filling the questionnaire was reported by 81% of the participants.

Among 31 recent users of pesticides, 14 reported that they did not know which type of pesticide they had used. The other 17 persons reported the recent use of one to four different products. For some of these products, only the brand name of a local distributor was recalled and did not allow for the identification of the active ingredient. The identifiable pesticides included herbicides, fungicides, and insecticides. Among herbicides, the organophosphorous compound glyphosate was mentioned four times, in addition to a combination of flumioxazin and prodiamine, diquat, and paraquat dichloride (one single report each). Four different fungicidal formulations were reported: thiabendazole (three times), imazalil, aluminum-potassium-sulfate, and mancozeb as a combination of the two dithiocarbamates maneb and zineb (one reporting each). Among insecticides chlorpyrifos was mentioned three times and paraquat, ethoprop, and carbofuran were all mentioned once.

Two thirds of the pesticide users prepared the mixtures themselves. Only 3 of 31 farm workers that had used pesticides recently reported using gloves and only 6 reported using masks during active spraying. The main reason (67.7%) stated for not using protective equipment was that masks and gloves were not available. Other reasons provided were “uncomfortable” (9.7%) and “not necessary” (12.9%). Reasons were not provided by 9.7%.

About 61% of the pesticide users washed their hands when still on the field. Moreover, 61% changed clothes immediately after work. Spraying equipment was not stored inside the home. Equipment was cleaned outside the garden/yard in about 90%. Three persons stated that they clean the equipment after use in a nearby river or creek. More than 70% disposed of left-over pesticides in the garden or in a river. Empty containers were not used for other purposes, e.g., storing food. About 71% of farm workers lived more than 1 km away from the plantation where they work.

Aerial spraying is a very common form of pesticide application in conventional farming of Ecuador. To assess the exposure linked to this form of application, the participants of both groups were asked how often they observed aerial spraying either directly above or near their dwelling. As shown in Table 3, pesticide users were exposed more often to aerial pesticide spraying. The participants were further questioned for noticeable pesticide residues after this aerial spraying, in the form of smell or moisture on the skin or clothing.

All farm workers stated not keeping any pesticide canisters or spraying equipment in their homes. Required cleaning of spraying equipment was performed outside the own yard or garden by 90% of the respondents and three farmworkers stated that they wash the equipment in a nearby natural water body (creek, river). Most plantation workers reported disposal of residue pesticides in their garden or in a nearby river. Use of empty pesticide containers for storage of food or other purposes was not reported.

There was a clear difference between the two groups in regard of their personal assessment of the harmfulness of pesticide use as well as the impact of pesticides for the environment (Table 4), but almost all participants answered the question if spraying of pesticides may be dangerous for their health and the environment in the affirmative (except 2 of 68).

When asked for their reason for using and spraying pesticides, the majority (70.9%) stated that the instructions had been given by a superior, 42.2% that the treatment is good for the plants, 41.9% that pesticides reduce the overall effort, and 38.7% that the spraying will result in a higher yield. Of the participants, 39% declared to be ready to stop using pesticides if would not affect their income negatively.

Non-pesticide users also indicated more knowledge on alternatives to intensive pesticide use, like the use of bio-pesticides or organic farming. Both groups reported little knowledge about the use of crop rotation and intercropping for keeping soil fertility.

### 3.3. Symptoms

Symptoms of local irritation (skin, eyes, and gut) and systemic symptoms like fatigue, nausea and dizziness were considerably more frequent in workers exposed to pesticides (Table 5). A more detailed analysis of the impact of aerial spraying was performed, as this spatially distributed form of application may affect the health of conventional as well as ecological farmworkers. Indeed, the observation of aerial spraying and subsequent related perceptions (smell, moisture on skin) were found to be with high significance associated to acute symptoms like dizziness, nausea, vomiting, and sleeplessness. Symptoms like strong fatigue, stomach pain, skin irritation, rashes, and watering eyes were also significant at the 0.05 level. This indicates that both conventional and ecological farmworkers are affected by aerial spraying. This exposure path reduces the health related differences between the two groups of farmworkers as could be expected in comparison to completely unexposed controls.

## 4. Discussion

The aim of the study was to examine possible relationships between indicators of exposure and health effects with regard to the different banana cultivation methods. A questionnaire survey, as applied in this study, is better suited for examining associations between current or very recent (last six month) health problems and exposure. It is less efficient for the study of long-term health effects and past effects. Therefore, we are confident that our results about frequency of recent symptoms do not suffer from serious recall bias, especially since the participants themselves did not voice very strong concerns about individual health effects of pesticides and also since the pattern of symptoms reported is overall plausible.

A total of 71 farm workers with a mean age of 45/46 years participated in the study. With regard to physiological attributes, there were practically no differences between the two groups. In the statistical analysis (logistic regression), differences in socio-demographic features (educational level, age) were controlled for. While age did not affect reported symptoms frequency significantly, a few symptoms were reported less frequently by the better educated workers. The effect of education was significant for dizziness, excessive salivation, watering eyes, and skin irritation, but the latter was no longer significant after controlling for the exposure group. The two groups differed considerably with regard to pesticide exposure (both in terms of own application and exposure through aerial spraying). This is an essential prerequisite for examining possible group-specific differences, for example in terms of the occurrence of symptoms.

Assessment of exposure to pesticides proved to be a special challenge for the study. Pesticide users are exposed to biocides through two routes, contamination during occupational application and aerial spraying. However, non-pesticide users are also affected by aerial spraying (through drift). This overlap in exposure may reduce differences between the groups with regard to health symptoms. Our analysis showed that pesticide impacts (moisture on the skin or smell) are perceived considerably more often by pesticide users than by non-pesticide users. Therefore, the differentiation of effects from occupational and from “environmental” exposure is not straightforward.

Aerial spraying because of pesticide drift, etc. is a cause of severe health concerns and therefore should be restricted or banned and only allowed in exceptional cases. At least, if the area to be sprayed is in close proximity to areas open to the public, specific risk management measures should be included in the approval to prevent adverse effects on the health of bystanders. For example, in the European Union, the area to be sprayed may not be in close proximity to residential areas [36]. However, in the regions of the present study, such precautionary measures are not common practice.

The participants were asked about health symptoms experienced in the last six months. The results demonstrate significant differences between the two groups: local irritation symptoms as well as systemic effects were considerably more frequent in pesticide users. Pesticide users had a six- to almost eight-fold increased risk for reporting gastrointestinal symptoms (mostly nausea, vomiting, and diarrhea) than non-pesticide users. Furthermore, symptoms of local irritation of skin and eyes were four-fold more prevalent and systemic neuro-vegetative symptoms like dizziness and fatigue were reported about five times more frequently. Irregular heartbeat (with nearly six-fold increased risk) could also be grouped into the same class of symptoms. Recall of symptoms during recent months should be pretty straightforward preventing relevant recall bias. The questions about symptoms were posed in a standardized and non-suggestive way making differential reporting bias unlikely. Unadjusted and adjusted odds ratios did not differ substantially nor systematically rendering relevant residual confounding unlikely. Thus the findings indicate that the use of pesticides is associated with acute adverse health effects in farm workers.

### 4.1. Pesticides Used

The extensive use of pesticides in conventional farming, in particular in developing countries, is well documented [37]. Among the active substances there are also products that are already banned or soon to be banned in the European Union. An example is paraquat (Gramoxon©), mentioned by pesticide users in existing studies [38] and banned in the EU since 2007 [39].

Pesticides applied by farm workers in conventional agriculture include chemicals suspected to be carcinogenic, in first line glyphosate (Roundup©). The International Agency for Research on Cancer (IARC) has classified glyphosate as a substance of Group 2A (probably carcinogenic to humans) [40,41]. Not less than 8 of 17 farm workers indicating specific pesticides said that they are using this herbicide. Also frequently mentioned by the farmworkers was ethoprop, another highly toxic pesticide belonging to the group of organophosphates and classified as probably carcinogenic to humans by the U.S. Environmental Protection Agency [42]. Our study thus provides additional confirmation that substances hazardous to health are used in conventional farming, and are usually applied without sufficient measures to protect the workers.

A cause for concern is the high number of persons with little knowledge about the specific pesticides they were applying. This may be based either on true ignorance or on the reluctance of participants to provide any information in this regard (various apprehensions).

Due to the relatively low number of persons (*n* = 14) who indicated which pesticides they were applying/using, a statistical analysis of possible relationships between pesticides used and health symptoms experienced is not possible. No information could be obtained on substances used for aerial spraying. Workers in conventional farming are exposed when mixing and applying pesticides but also when working in the sprayed fields no matter whether applied from the ground or from the air. Therefore, it is not possible to relate the symptoms to specific pesticides. Symptoms of central nervous system disruption would typically be associated with the reported organophosphate and carbamate insecticides, but also sensitization to pesticides among banana plantation workers is a frequent occupational health problem [43,44], maybe explaining the high odds ratios for skin problems.

### 4.2. Protection Measures

Uptake of pesticides in occupational exposure may occur in particular during mixing and applying/spraying. Organophosphates, for example, are absorbed through the skin and the respiratory tract. Therefore, from the perspective of occupational medicine, first priority should be given to (simple) measures to reduce exposure, apart from using less toxic products. Such measures include appropriate personal equipment to protect the respiratory organs, eyes, and hands.

Although almost all pesticide users surveyed acknowledge that pesticides are harmful to health, only about 20 percent of the respondents always use masks and/or gloves. A main reason for this inadequate use of personal protective measures is that masks and gloves are not available and/or not provided by employers. As to why this happens, two reasons are very likely: either ignorance or denial of the health risks involved or a reluctance of employers to provide this safety equipment out of organizational or financial considerations, though they have an obligation to do so. In any case, such deficiencies are reported frequently from countries of the Global South [45,46,47].

### 4.3. Comparison with Our Previous Study

In our previous study on pesticide impacts on coffee farm workers in the Dominican Republic (D.R.), we found an increased frequency of chromosomal damage in buccal mucosa cells [48,49] and also an increased symptoms rate in the exposed group [32]. When we presented the first results of our D.R. study, we were invited to participate in the Ecuador study as well. This provided an opportunity to confirm the original results in different settings. In a first paper [33], we already confirmed the higher frequency of cytological anomalies in buccal cells of workers exposed to pesticides compared to controls. This paper was in agreement with our paper about workers from the D.R. With the current paper, we also replicated the findings regarding symptom rates. In the current study, additionally, very high odds ratios were observed for symptoms of skin irritation. This might be typical for workers in banana plantations where often high amounts of fungicides like chlorothalonil are used [50], but also other symptoms and diseases are associated with aerial spraying of fungicides [51]. In another article, we reported that pesticide workers sired fewer children, especially when their pesticide exposure started at young age [52]. Interestingly, also in our current study, the pesticide workers reported (insignificantly) fewer children on average but the overall number of children was lower in the participants from Ecuador than from the D.R., maybe due to cultural influences. Furthermore, because of the smaller number of participants and the lack of plausible information regarding age at first exposure we could not investigate that issue further.

## 5. Conclusions

A reduction in the use of pesticides, i.e., switching to natural cultivation methods, would improve the health status of the local farming population, the environmental status of agricultural regions and the quality of the products. Natural cultivation methods might also seem a risk factor for farmers, e.g., via drop in yields, pointing out the importance of fair pricing policies. Yet, attempts and suggestions on a more sustainable, but still profitable production have been made for banana production [3], and a recent review shows that overall a large number of farmers benefit economically from sustainability standards [53]. Yet, there is substantial heterogeneity regarding benefits, showing that several more factors play a decisive role [53], e.g., the organization of supply chains. However, limited to cultivation methods, there are also aspects that can be considered in a narrower context. Such cultivation methods would include, among others, appropriate plant nutrition and soil fertility management combined with crop rotation [54], appropriate irrigation management, and timing of sowing or planting in order to reduce pests. Precision farming like spraying of hot-spots and weeding with optical detectors has also been demonstrated to be effective in banana cultivation [55]. Intercropping (when it is possible) and the use of variety mixtures limit the spread of pests and diseases and provide food and shelter for natural enemies of pests. Bio-control and natural pesticides are other options [56] as well as integrated pest management (IPM) [57]. Exerting more pressure on consumer market supply chains and thus on the employers of farm workers may also contribute to an improvement of the health-threatening working conditions (improved safety measures, a reduced or no use of pesticides). Representative population surveys show that there is a strong public interest in the issue “pesticides in food” in Europe (EFSA Special Eurobarometer Food Safety in the EU). Pesticide residues in food are generally perceived as a health risk. Systematic reviews demonstrate that products from environmentally responsible agricultural methods contain considerably less hazardous substances than conventional food products. This could be a starting point to significantly raise awareness of working conditions in banana producing regions. Fair international trade agreements and better labor right enforcement mechanisms could also lead to an improvement of the situation.

## Figures and Tables

**Table 1 ijerph-18-01126-t001:** Overview of survey locations and number of participants.

Survey Location	Number of Participants	Date of Examination
Quevedo	10	26 October 2015
La Union	7	27 October 2015
Valencia	17	28 October 2015
La Libertad	23	29 October 2015
Buenavista	14	30 October 2015

**Table 2 ijerph-18-01126-t002:** Socio-demographic and anthropometric data by pesticide use.

	Pesticide Users	Non-Users	*p*-Value ^1^
Age (years)	45.9 ± 13.4	44.7 ± 16.6	0.748
Height (cm)	164.9 ± 4.8	165.6 ± 5.6	0.594
Weight (kg)	69.4 ± 10.7	69.6 ± 11.2	0.940
Number of own children	2.8 ± 2.3	3.1 ± 2.2	0.616
Number of persons in household	4.4 ± 2.0	4.1 ± 1.6	0.484
Father working in agriculture	80.7%	78.4%	0.818
Mother working in agriculture	35.5%	54.1%	0.124
Education			**0.036**
None	6	1	
Compulsory	22	22	
Secondary	6	14	

^1^ Bold: *p*-value < 0.05.

**Table 3 ijerph-18-01126-t003:** Frequency of exposure to aerial spraying and reported residues after spraying (in %).

	Pesticide Users	Non-Users	*p*-Value ^1^
**Aerial spraying observed**			**0.001**
Never	0.0	24.3	
Once per month	16.6	40.6	
Once per week	54.9	0.0	
More than once per week	25.8	0.0	
Daily	3.2	10.8	
**Perception (smell, moisture)**			**<0.001**
Never	0.0	27.6	
In less than half of the cases	25.8	6.9	
In more than half of the cases	0.0	10.3	
Always	74.2	55.2	

^1^ Bold: *p*-value < 0.05.

**Table 4 ijerph-18-01126-t004:** Knowledge and perceptions about pesticides (in %).

	Pesticide Users	Non-Users	*p*-Value ^1^
**Assessment**			**<0.001**
Not harmful	9.7	5.4	
Moderately harmful	90.3	16.2	
Very harmful	0.0	78.4	
**Alternatives**			
Biopesticides	3.2	40.5	**<0.001**
Organic farming	25.8	100	**<0.001**
Crop rotation/sequencing	11.9	29.7	0.089
Cultivating crop mixtures	29.0	21.6	0.482

^1^ Bold: *p*-value < 0.05.

**Table 5 ijerph-18-01126-t005:** Difference in reported symptom rates by group.

Symptom	Unadjusted	Adjusted ^1^ Logistic Regression		
	OR ^2^	OR	95% CI	*p*-Value ^2^
Headache	1.49	1.47	0.54–4.05	0.453
Vision problems	0.83	0.79	0.28–2.18	0.643
Dizziness	4.44	4.80	1.55–14.87	**0.007**
Nausea, vomiting	7.93	7.50	1.77–31.77	**0.006**
Excess salivation	2.36	1.82	0.61–5.39	0.281
Strong fatigue	4.36	4.96	1.65–14.88	**0.004**
Exhaustion	**3.27**	2.53	0.88–7.28	0.086
Stomach pain	2.13	2.22	0.76–6.53	0.147
Diarrhea	4.08	6.43	1.06–39.00	**0.043**
Sleeplessness	2.70	3.39	1.16–9.87	**0.025**
Burning eyes	3.82	4.10	1.37–12.31	**0.012**
Skin irritations	4.59	3.58	1.10–11.71	**0.035**
Runny nose	2.67	2.79	0.77–10.11	0.119
Breathing difficulties	3.06	2.83	0.80–9.99	0.105
Irregular heartbeat	7.29	5.75	1.08–30.67	**0.041**
Watering eyes	**3.63**	3.12	0.98–9.95	0.055
Skin rashes	3.49	3.38	0.71–16.11	0.126
Cough	1.98	2.10	0.66–6.67	0.209
Twitches, trembling	2.33	3.58	0.52–24.61	0.195

^1^ Adjusted for age and school education; ^2^ Bold: *p*-value < 0.05.

## Data Availability

The data presented in this study are available on request from the corresponding author.

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
