# Peer review of "Health Symptoms Related to Pesticide Use in Farmers and Laborers of Ecological and Conventional Banana Plantations in Ecuador"

_ijerph, 2021, doi:10.3390/ijerph18031126_

Round 1

Reviewer 1 Report

The paper presents a study of a pressing environmental and occupational health issue in Ecuador. It compares self-reported symptoms in pesticide-exposed individuals working in conventional banana production to those of independent agro-ecological banana producers. While this is important, the paper’s major deficiency is the identification of a substantial contribution to knowledge. Finding that workers who are exposed to pesticides report more symptoms, but also rate pesticides as less hazardous, is not actually new. The authors need to think about what this study actually contributes to knowledge on the subject. Why is it necessary to publish/read in terms of a novel contribution? If the reason is to put pressure on supermarkets or other stakeholders who could improve the situation of workers and farmers in Ecuador, the authors need to consider whether a cross-sectional study of self-reported symptoms will really do the trick.

I therefore recommend major revisions, both to better identify and then communicate a significant contribution, and to generally improve the quality of writing throughout. I also have some suggestions regarding providing more methodological detail and reflection on possible forms of epidemiologic bias; how to frame the study in terms of geographic regions such as ‘tropical’ countries, ‘the Global South’, etc.; improving the overall framing in terms of the structural causes of pesticide exposure; and revisiting the practice recommendations. More detailed comments are provided below, section-by-section.

Abstract

  • “Results of the Buccal Micronucleus Cytome Assay have been reported in a previous paper.” I don’t think this belongs in an abstract.
  • “Significance was reached in 8 out of 19 investigated 19 symptoms. This concerned symptoms of local irritation like skin and eye irritation” This is confusingly written, as it’s not clear what ‘this’ refers to. In general, the quality of writing in the abstract needs to be improved as it doesn’t adequately communicate the important aspects of the study.
  • I suggest reporting confidence intervals with odds ratios.

Introduction

  • The Introduction needs to do a better job of explaining the study’s contribution, both in terms of why the general topic is important, and why these specific results add to existing knowledge on it in an important way. The organization of the introduction could also accomplish such an explanation more efficiently and less confusingly. While I don’t want to be overly prescriptive, one efficient Intro structure for a scientific paper starts by establishing the importance of the general topic area (as you do), then reviews literature on what’s known about it and identifies a ‘gap’ in that knowledge before explaining how the paper being introduced helps fill that gap. Right now the paper throws some information about the specific study into the middle of the literature review, inexplicably and jarringly. I had trouble identifying a specific contribution being made by the present paper.
  • The paper currently appears to be contributing to literature on pesticide problems in places where “tropical ‘cash crops’” are grown, but the specific characteristics of such places are confusingly presented. The reference to “products that are long forbidden to use in industrial countries” suggests that the geographic region in question does not contain ‘industrial countries’ and some generalizations are made about PPE availability, container labelling and pesticide handling by citing a very general WHO source (lines 34-36 on p1). The review of literature needs to avoid overgeneralizing about pesticide usage and impacts in some vaguely-defined, ‘tropical’ and not-‘industrial’ part of the world. For example, you note the “generally poor economic situation in most of the affected regions” and substantiate the point with a study about potato farmers in Uganda. I strongly urge you to improve your geographic specificity about countries where bananas are produced and avoid lumping them into an amorphous ‘tropical’, ‘developing country’, ‘global South’ or ‘Third World’ region that also contains countries in sub-Saharan Africa (and their potato farmers). It may be appropriate or helpful to generalize about banana-producing countries using such terms (and global South is likely the least offensive term of the ones I’ve mentioned) but you should explain exactly what your chosen term means and – specifically – how it is relevant to both pesticide exposures and the current situation in Ecuador.
  • Similarly, the review of literature cites sources discussing pesticide-related impacts among other crops than bananas (which have different mixes of chemicals in use), in other parts of the world, or from earlier eras of banana production (e.g. Wesseling et al., 1996) when the mix of chemicals in use was significantly different. If you are going to review literature on pesticide impacts in other places, you need to explain how those results are or aren’t relevant to the study you are conducting – on a specific mixture of pesticides in use with a specific banana-growing population in a specific banana-producing South American country, at the end of the second decade of the 21st You do have material explaining the complexities of determining causality, but I don’t currently see your study as helping to fill that gap.
  • “Banana farming is a very representative sector for studies of pesticide related health effects, as a large amount of these substances gets applied in conventional agriculture.” It’s true that banana production is an illustrative sector and uses a lot of pesticides, but the specific mix of chemicals in use (i.e. large amounts of fungicides applied aerially) and production methods – often quite large plantations with precarious workforces – makes the situation there not exactly ‘representative’ of pesticide use in other sectors such as (for example) potatoes or non-traditional export crops.
  • “Due to lack of regulation, education and control of the use of pesticides…” This passage suggests that Ecuador suffers from these things. However, there are regulations on pesticide use in Ecuador, but they’re not adequately observed or enforced. I suggest you get more precise about how you characterize the country, and the region of the world you say it represents.
  • “Symptoms of acute pesticide poisoning in banana workers are reported quite often [13,14].” The cited sources are one WHO fact sheet and one study done with farmers in Jamaica where it is likely that banana production is done differently (I suspect by more small-scale, pesticide-using farmers). You should be more specific about experiences of ‘banana workers’. Acute pesticide poisoning in banana workers appears to have declined since organophosphate and carbamate nematicides fell out of favour. In any case, you should explain how the populations in the sources you are citing do, or do not, reflect the situation you are currently studying in Ecuador. This might also help you to better explain your paper’s original contribution.
  • “We already presented our results of a cross-sectional medical survey in Ecuador regarding 61 indicators of genotoxicity in conventional and ecological farming [16] and provide here comparative results, focusing on self-reported symptoms in conventional and ecological banana farmworkers that may be attributed to pesticide ” This sentence tells me you’ve already published from this study, but doesn’t tell me what you found, which would be an obvious, important component of existing scientific knowledge on exactly your topic. It would also help to identify a novel contribution that this study is making, because right now I don’t see one. This sentence suggests that the importance is in providing a comparison to your recently-published study on genotoxic effects in the same population, but this isn’t obviously an important contribution to knowledge. The conclusion that banana workers who are exposed to pesticides report more pesticide-related symptoms is important but not really news, scientifically speaking. Why was it important to publish this study? Of course it’s important to reduce pesticide exposures and promote agro-ecological farming, but you’ve submitted your findings to a scientific journal, so should clarify what’s scientifically interesting/novel about them. Maybe reporting more about what you found with the buccal cell analyses would clarify how this study adds to them.
  • There are some other studies you could review on pesticide use and health in banana production, including in Ecuador. Studies by Wesseling et al. (in addition to those cited), Penagos, Breilh, Maldonado and Harari et al. are all relevant quantitative examples, while Polo Almeida’s work in Tenguel is also worth reading and possibly citing. It is important when doing research in Ecuador to read and cite the work of Ecuadorian researchers (e.g. Breilh, Maldonado, Harari, Polo Almeida, including in Spanish) to avoid ‘academic imperialism’.

Materials and Methods

  • The study sites don’t really represent a ‘cross-section of Equador’ (sic). They cover two main parts of the banana-producing coastal region, which is not the same as a cross-section of the entire country.
  • Since selection of participants was not random, it is important to discuss the possibility for selection bias and any possible implications for the results.
  • More information about the questionnaire that was used and the ‘standard form’ it was developed from would be helpful. There is major potential for information bias in this particular study design (i.e. working with people who know that pesticides are harmful, and their own exposure profile) and it’s important to discuss how you accounted for that possibility.
  • It’s unclear if interviewers were aware of exposure status. I assume that’s what the anonymization was for, but this should be clearer. Also, since the two groups differed significantly in terms of education, it might have been possible for the interviewers to notice patterns and tell which group they were talking to. How did you account for possible interviewer bias?
  • “Categorical data for the groups of pesticide users and non-users were compared by Chi-squared test of Fisher's exact probability test for binomial categories.” I assume you mean ‘Chi-squared test or Fisher’s exact…’?

 Results

  • Title has a typo: “3.1. Socio-democraphics and education of the participants” (‘socio-demographics’)
  • “school experience. Accordingly, 14 persons from the ecological group had attended a higher school…” ‘a higher school’ is not that meaningful in English. Maybe say ‘secondary school’?
  • “in terms of current u and lifetime use of pesticides”…what is the ‘u’ doing in this sentence?
  • “Herbicides were in most cases organophosphates…” Which ones? A table listing the kinds of pesticides that were reported would be helpful, even if you have to include it as a Supplementary file.
  • “In the group of used fungicides…” This passage doesn’t make sense.
  • The statement that organophosphates were the most commonly-applied pesticide (8/31 workers) doesn’t account for the fact that aerial fumigations with (non-organophosphate) fungicides may cause significant exposure to more, if not all, of the conventional workers who wouldn’t apply these themselves (since they’re not pilots) but would be exposed in the fields. This should be discussed.
  • “residue” has a specific meaning in relation to pesticides, i.e. residues on imported produce that pose a risk to consumers (etc.); I suggest using something with a less ambiguous meaning, like ‘leftover pesticides’.

Table 2

  • This table isn’t self-explanatory. You should explain better than ‘father farmer’ and ‘mother farmer’.

Table 3

  • Why are p-values only listed for some of the differences in frequencies? You could include all the p-values but put an asterisk or use bold type to indicate significant differences.

Table 5

  • Are the odds ratios in Table 5 for the full logistic regression model, adjusted for “age and school education covariates” (I assume yes, as mentioned in the Discussion)? I would like to see the odds ratios for these covariates and any others in the model. There should be room if you make a bit better use of space in this table. If education (a proxy for class or socio-economic status) explains much of the observed differences, this would be an important finding to note and attempt to explain.

Discussion

  • “It has to be noted that aerial spraying is banned in the European Union since 2009 (because of pesticide drift etc.), with a deadline for implementation by 2011, and only allowed in exceptional cases.” Why does this ‘have to be noted’? The relevance of the EU regulation to Ecuador is not clear. If the point is that aerial spraying causes exposure via drift, this can be noted without discussing the EU situation.
  • “The participants were asked about any health symptoms experienced in the last six months. The results demonstrate significant differences between the two groups: Both local irritation symptoms and systemic effects were considerably more frequent in pesticide users. This indicates that the use of pesticides is associated with acute adverse health effects in farm workers.” This interpretation is a bit too bold in light of the study design. The findings don’t necessarily indicate an association with ‘acute adverse health effects’ but rather ‘reporting acute adverse health effects’. The authors should be a bit more circumspect about what they have actually found with their study and what it means. A discussion of possible information and selection bias would help in reaching a good level of confidence / tentativeness regarding what the findings mean.
  • Comparing your results to findings about the “extensive use of pesticides in conventional farming, in particular in the Global South” (as noted in a 19-year-old source about ‘pesticide use in developing countries’) is not really appropriate. First of all, Ecobichon’s paper is not about ‘the Global South’ or at least he doesn’t use that expression; it is not up-to-date; and it makes enormous generalizations about ‘developing countries’. I strongly urge you to avoid presenting Ecuador as some sort of mysteriously or inexplicably ‘poor’, ‘Southern’, ‘developing’ or ‘tropical’ country, and make at least an effort to explain the poverty that structures both banana production and hazardous pesticide use in Ecuador as a result of colonial and ongoing neocolonial exploitation.
  • The reference to paraquat in the Discussion cites a study done in Colombia. This herbicide was ‘restricted’ in Ecuador in 2018 (https://www.agrocalidad.gob.ec/wp-content/uploads/2020/05/ny6.pdf). Did workers in your study actually say they were still applying paraquat? I would clarify whether it is still in use in Ecuador at all, and – if so – under what conditions. This would be worth discussing in the Discussion.
  • “A reduction in the use of pesticides, i.e. switching to natural cultivation methods, would improve the health status of the local farming population, the environmental status of agricultural regions and the quality of the products. This is in line with the objective of protecting health and safety in Austria.” The reference to Austria does not make sense here. I understand that the authors are concerned about health and safety in Austria, but the paper is not about that, nor can the readership be assumed to care enough to single out that country.
  • The recommendations for action (putting pressure on supermarkets) do not go near the root causes of hazardous working conditions in Ecuador in the form of North-South inequities and the power relations that maintain them, both between Ecuador and banana-consuming countries, and between banana workers and the local economic elites who control banana production in Ecuador. While I understand that the authors are not social or political scientists, I urge them to read more about the history and political economy of banana production, to avoid proposing solutions that leave Northern consumers in the driver’s seat and thus maintain colonial inequities. This is outside the scope of many health science papers but I would like to see at least an attempt, reflecting some additional background reading.

General writing comments

  • In general, the quality of writing is mediocre to poor. I see that the same research team’s 2020 paper in this journal is much better written. While it is of course unfair and counterproductive that English has become the dominant language of academic communication, it is important that results be communicated well in whichever language is being employed. The high quality of writing in the team’s already-published paper on the same study indicates that they have the capacity to write better in English; I strongly suggest they draw on this capacity and bring the paper up to an acceptable level before resubmitting. Some examples of phrases that are confusing or poorly written are provided below as illustration of the need for improvement in the writing quality, but these are only a few of many possible examples. The entire paper’s writing quality needs improvement, and not just these passages:
    • “Application equipment may be of poor quality in terms of handling security.” What does ‘in terms of handling security’ mean?
    • “we performed a project comprising several studies to obtain information about burden and health outcomes in pesticide using and ecological farmworkers of Ecuador at a scientific level” Why is it necessary or meaningful to explain that the studies were ‘at a scientific level’?
    • “Due to the labor intensive related tasks, a significant section of the local population gets exposed, either directly or indirectly.” What’s a ‘labor intensive related task’ and how does it lead to a ‘section’ (I suggest ‘segment’ as a better word choice) being exposed?
    • Make sure you spell Ecuador correctly throughout (it’s not Equador).
    • “Three persons stated to clean the equipment in a nearby river or creek. More than 70 percent disposed pesticides residues in the garden or in a river.” The first of these sentences has an incorrect and misleading sentence structure. The second uses the term ‘pesticide residue’ where you should probably say something like ‘leftover pesticides’. ‘Pesticide residues’ are generally understood to be trace amounts of pesticides that arrive at markets and represent an exposure for consumers.

Reviewer 2 Report

 Line 17-18: Need more clarification and information in results section. Which paper are you referring to?

Line 21-22:What do you mean majority? How many?

Line 46: Change "variable" to "vary"

Line 48-51: Do you have reference for this paragraph?

Line 77: The numbering "(3)" should be before "an". Otherwise I suggest you remove the numbers.

Line258: Highlight a few of natural cultivation methods you are suggesting.

Reviewer 3 Report

Dear Authors,

The subject described in the paper (ijerph-1041239) is interesting and significant, and the article can be submitted for publication after revision.

Comments to the article:
- the introduction part can be improved.

- the discussion part can be improved.

They are concise. The literature is broad (e.g., about the situation concerning pesticide use at banana plantations in other countries).

- Moderate English corrections are required. Only some examples are presented below: 
- line 43-44, the text may be improved in such a way:
Affected persons may get exposed by direct handling and application of pesticides, as bystanders, by entering pesticide-treated fields, cleaning and storing equipment, indirect routes, and contamination of water, food, or clothing [7-9].
-line 52-53, the text may be improved in such a way:
Banana farming is a very representative sector for studies of pesticide-related health effects, as large amounts of these substances are applied in conventional agriculture.
- line 52-53, the text may be improved in such a way:
The generally poor economic situation in most affected regions results in a lack of regulative measures and law enforcement regarding pesticide use. It is also linked to little awareness and a risk-taking attitude of workers [15].
-line 58-59-60 the text may be improved in such a way:
The latter is a consequence of their daily fight for a minimal quality of life than knowingly accepting conditions for compromised personal health.

Apart from corrections, improving correctness and style, minor spell checks are required.

line 68 The map would help readers to understand where the study was performed or it is possible to direct readers to another article where the map was presented.

- tables 3,4,5 - some results are bolded - please add an explanation.

Kind regards

Round 2

Reviewer 1 Report

The paper has been improved in terms of methodological detail and citation of relevant literature. The writing is better, but still needs work as discussed below and in the PDF. In addition, the paper’s contribution is still not clear. These things might be fix-able with just minor revisions at this point.

You still haven’t identified an important contribution to knowledge in the introduction. Influencing a European policy decision is important, but not grounds for a novel contribution to scientific knowledge. What does this study contribute that people don’t already know about pesticide exposures in banana production or in export-based agriculture of tropical crops more generally (or some other important public health research question)? You could explain that the mix of pesticides used in banana production changes over time so exploratory designs like cross-sectional studies of self-reported symptoms remain relevant as a way to try to keep up with the changing possible impacts...or some other scientifically interesting contribution (which you as the research team should know better than I do, since you planned and did the research). Of course, the contribution to knowledge then needs to be discussed in the Discussion.

The addition of mention of ‘structural’ factors and the reference to Polo Almeida is good, but its meaning isn’t clear. At least a half-sentence of what such structural factors are and how they shape patterns of exposure is necessary. A good explanation of how those structural factors operate in coastal Ecuador might also enhance your policy recommendations.

The writing still needs work, both in the newly-added material and – frustratingly – in some problems that I noted in my first review that were still not fixed. I have put highlights and comments in the PDF I reviewed with some suggestions on improving the communication in English. Please actually make these changes, so as to not waste the effort and goodwill of peers who volunteer their time to review your work. I note that there is a whole paragraph on page 4 (lines 137-146) that is completely new, and also in noticeably better-quality English than the original submission. Please bring the whole article up to this standard, even if it requires more of a time commitment from the member(s) of the research team with advanced English skills. The quality of the paper will otherwise be compromised, and peer reviewers will also have to spend their time doing copy-editing work that should be done by the research team.

There are under- or inappropriately substantiated points that cite references that do not adequately support them. For example, you still cite one study of Ugandan potato farmers to make an enormous generalization about pesticide use in huge parts of the world. You also cite a study done in Ecuador’s Loja province to support a point about exposures of banana workers, but this study does not talk about actual banana workers on farms that produce for export. Ensure that your points are adequately substantiated.

There is one sentence in the Discussion saying that the temporal dimension reduces the risk of recall bias. This is still not an adequate discussion of the possibility of information bias in a self-reported symptoms study topic.

Explain what ‘systemic neuro-vegetative symptoms’ (p. 7, line 237) corresponds to in the findings reported in Table 5. This is not obvious.

You could do better in explaining your results in terms of reasons why PPE and other protective practices (precise aerial fumigations, observation of re-entry periods, bathroom breaks to wash hands, being allowed to leave the field during fumigations, etc.) are not always observed/possible in the study region. This would also help you do less generalizing about ‘the Global South’ or ‘developing countries’, as these generalizations don’t actually explain anything in terms of specific determinants of exposure. See comments in the PDF (and in my previous review, which you largely ignored).

The conclusion that switching to natural cultivation methods would improve health in the study area is too confident. Many banana farmers might see a drop in yields and therefore overall income without pesticides (at least given the constraints imposed by exporters, supermarkets, etc.). There is more to health than just pesticide exposure (although of course this is important), and you haven’t discussed other relevant factors such as social determinants of health. You need to hedge your conclusions and make them more cautious. The switch to organic production is also time- and resource-intensive and not practical for many farmers without external funding/support/credit. You should inform yourselves on the political and economic aspects of banana production and how they relate to health and its non-chemical determinants in order to refine your recommendations/ conclusions. I’m not saying that farmers should keep doing conventional, pesticide-intensive production – I’m just saying that the issue is more complicated than Section 5 makes it seem. You may get away with publishing this paper without going into those complications, but you will very likely be more effective in helping your Ecuadorian partners (and also more informed by highly relevant social determinants / determination of health scholarship) if you start engaging at the power relations that govern banana production and might complicate the changes you are advocating. Doing more with the source by Polo Almeida is one way to start moving in this direction.

Author Response

please see our detailed response in the attached file!
